# The Impact of Recirculation on Extracorporeal Gas Exchange and Patient Oxygenation during Veno-Venous Extracorporeal Membrane Oxygenation—Results of an Observational Clinical Trial

**DOI:** 10.3390/jcm12020416

**Published:** 2023-01-04

**Authors:** Johannes Gehron, Dirk Bandorski, Konstantin Mayer, Andreas Böning

**Affiliations:** 1Department of Adult and Pediatric Cardiovascular Surgery, University Hospital Giessen, 35392 Giessen, Germany; 2Faculty of Medicine, Semmelweis University Campus Hamburg, 20099 Hamburg, Germany; 3Medizinische Klinik 4, Pneumologie, Infektiologie und Schlafmedizin, ViDia, Christliche Kliniken Karlsruhe, 76137 Karlsruhe, Germany

**Keywords:** veno-venous extracorporeal membrane oxygenation, recirculation, ultrasound dilution, critical care, acute respiratory distress syndrome, circulatory and respiratory physiological phenomena

## Abstract

Background: Recirculation during veno-venous extracorporeal membrane oxygenation reduces extracorporeal oxygen exchange and patient oxygenation. To minimize recirculation and maximize oxygen delivery (DO_2_) the interaction of cannulation, ECMO flow and cardiac output requires careful consideration. We investigated this interaction in an observational trial. Methods: In 19 patients with acute respiratory distress syndrome and ECMO, we measured recirculation with the ultrasound dilution technique and calculated extracorporeal oxygen transfer (VO_2_), extracorporeal oxygen delivery (DO_2_) and patient oxygenation. To assess the impact of cardiac output (CO), we included CO measurement through pulse contour analysis. Results: In all patients, there was a median recirculation rate of approximately 14–16%, with a maximum rate of 58%. Recirculation rates >35% occurred in 13–14% of all cases. In contrast to decreasing extracorporeal gas exchange with increasing ECMO flow and recirculation, patient oxygenation increased with greater ECMO flows. High CO diminished recirculation by between 5–20%. Conclusions: Extracorporeal gas exchange masks the importance of DO_2_ and its effects on patients. We assume that increasing DO_2_ is more important than reduced VO_2_. A negative correlation of recirculation to CO adds to the complexity of this phenomenon. Patient oxygenation may be optimized with the direct measurement of recirculation.

## 1. Introduction

Veno-venous extracorporeal membrane oxygenation (VV-ECMO) is an important option in the treatment of patients suffering from severe forms of acute respiratory distress syndrome (ARDS). It is known to allow pulmonary recovery based on extracorporeal gas exchange [1]. Although thought to completely replace the native pulmonary gas exchange, the interaction between cannulation, ECMO blood flow (Q_ECMO_) and cardiac output (CO) affects the efficiency of extracorporeal gas exchange, as well as limits delivered oxygen (DO_2_) and patient arterial oxygenation [2,3,4].

Reduced extracorporeal gas exchange efficiency is initiated by the phenomenon of recirculation, which causes a hardly predictable return of extracorporeal oxygenated blood into the ECMO system because of the position of both ECMO cannulas [5]. The return of oxygenated blood thus decreases effective extracorporeal blood flow. Several determinants known to influence both recirculation and extracorporeal gas exchange have been described [6,7]. The position of the cannulas is one major determinant of the recirculation fraction (REC): a closer cannula position would increase recirculation, while a broader position would decrease the recirculation fraction. Despite the fixed cannula position, recirculation may be further affected by the native cardiac output of the patients and show two different results. A cardiac output close to the extracorporeal blood flow with a low CO/Q_ECMO_ ratio will increase recirculation, and a cardiac output higher than extracorporeal blood flow with a higher CO/Q_ECMO_ ratio will decrease recirculation. Both conditions reduce the fraction of venous blood oxygenated by the extracorporeal system and can cause systemic hypoxia.

As recirculation rises with increasing Q_ECMO_, it is thought that a corresponding higher recirculation reduces the oxygen delivery and leads to a further decrease of arterial patient oxygenation. To avoid this inefficient oxygen delivery with decreased patient oxygenation it was recommended to use a low Q_ECMO_ regime in clinical practice [6,8,9,10].

Several authors have attempted to estimate the aforementioned determinants to minimize recirculation. It may be required to either exclude high recirculation with blood gas analysis, to indirectly recalculate native venous saturation, or to directly measure recirculation with an appropriate device such as the thermodilution technique [9,10,11,12,13,14,15]. At least one of these determinants must be known to resolve the recirculation and its impact on patient oxygenation, otherwise only an approximation is possible during clinical routine. An estimation may not allow the discrimination of whether low patient arterial oxygenation may be caused by a close cannula position, an assumed inefficient oxygen delivery with high Q_ECMO_ or a patient-related high CO [16]. Systematic, but retrospective data, was provided by Zanella: he reported an increase in patient arterial oxygenation with increasing Q_ECMO_ and increasing REC with a mathematical model fitted to retrospective patient data [17]. Prospective clinical and systematic directly measured data beyond the mathematical approximations, and fitted data was so far not carried out. In our prospective clinical trial, we sought to address the questions of whether (1) higher blood flows and increased recirculation are automatically associated with an inefficient oxygen delivery and reduced arterial patient oxygenation and (2) if patient cardiac output affects this dynamic condition. To systematically investigate the determinants of recirculation, we conducted an observational study with patients suffering from ARDS and requiring VV-ECMO.

## 2. Materials and Methods

### 2.1. Study Design and Participants

The study was an observational and prospective clinical study that consecutively enrolled adult patients. They were eligible if, according to the Berlin definition of ARDS, they failed to reach acceptable blood oxygenation and decarboxylation levels with optimized ventilator settings, inspired O_2_ fraction ≥ 90%, peak airway pressure [30 cm H_2_O, inverse-ratio ventilation] and supportive treatment such as inhaled nitric oxide (NO), and required extracorporeal veno-venous ECMO support [18]. The study was approved by the local medical ethics committee (Justus Liebig University Giessen, 22/2012) and registered with the German Clinical Trials Register (DRKS, DRKS00005106). Due to its observational design, a control group was not required. Written informed consent was obtained from all patients or substitute decision makers for participation in the trial.

### 2.2. ECMO Management and Patient Interventions

We either set up a standard two-cannula femoro-jugular or double lumen cannulation approach for the patients with a standardized ECMO circuit. In all cases, the cannulation was guided by transoesophageal echocardiography (TEE). The sizes of the cannulas were in the range 19–31 F depending on the patients’ anatomical features. The ECMO equipment employed was a standard centrifugal pump-based system (PLS-Rotaflow; Maquet Cardiopulmonary AG, Rastatt, Germany) with a hollow-fiber oxygenator (Quadrox-D; Maquet Cardiopulmonary AG, Rastatt, Germany). Once optimal and stable, Q_ECMO_ and patient oxygenation and decarboxylation levels were discerned, and ventilation was gradually reduced for protective ventilation [19,20].

### 2.3. Patient Monitoring and Gas-Transfer Calculation

Besides standard ICU treatment, the patients received a CO-monitoring system based on arterial pulse contour analysis (Vigileo FloTrac™, Edwards, München, Germany) to be independent from right ventricle-based thermodilution principles [21]. Arterial and venous blood samples were drawn from the oxygenator during recirculation measurement to calculate ECMO oxygen transfer (VO_2 ECMO_). DO_2 ECMO_ was calculated from the blood gas results. To avoid extensive blood sampling, we utilized transcutaneous pulsoxymetry (SpO_2_) as a surrogate parameter for patient arterial saturation.

VO_2 ECMO_ was calculated according to the following equation: VO_2 ECMO_ = (mL O_2_ × min^−1^) = caO_2_ − cvO_2_ × Q_ECMO_
where cxO_2_—content of oxygen in arterial (a) and venous (v) blood samples (mL O_2_/100 mL blood); Q_ECMO_—ECMO flow (L × min^−1^).

DO_2 ECMO_ was calculated according to the following equation: DO_2 ECMO_ = (mL O_2_ × min^−1^) = caO_2_ × Q_ECMO_
where caO_2_—content of oxygen in arterial blood samples (mL O_2_/100 mL blood); Q_ECMO_ – ECMO flow (L × min^−1^).

### 2.4. Recirculation Measurement and Protocol

Recirculation was measured with an ultrasound-based flow dilution principle that makes use of two probes for ECMO draining and returning tubing (ELSA, Transonic Systems Inc., Ithaca, NY, USA) [11,12]. With an injection of a 20 mL NaCl 0.9% bolus, the device computes the REC from upstream- and downstream-derived flow dilution signals. Recirculation, gas exchange and SpO_2_ were measured daily during ECMO. To determine the impact of recirculation, a series of measurements was carried out for each patient. First, the ECMO flow was decreased before the measurement until SpO_2_ reached a clinically accepted and safe minimal value of approximately 85% [22]. Then, the ECMO flow was gradually increased with 300–500 mL steps until a maximum stable ECMO flow without any fluctuation was accomplished. During each step, recirculation was assessed with corresponding VO_2 ECMO_, SpO_2_ and CO.

### 2.5. Outcome Measures

The primary outcome was to explore the impact of REC on patient oxygenation and VO_2 ECMO_ at increasing Q_ECMO_. We evaluated the clinical relevance of recirculation through investigating whether increasing ECMO flow causes higher recirculation and if it is associated with lower VO_2 ECMO_ and SpO_2_. Secondary endpoints were the oxygenator gas exchange (DO_2_, and VO_2 ECMO_), REC, ECMO flow (Q_ECMO_) and patient CO.

### 2.6. Data Collection and Statistical Analysis

As we did not expect a particular probability distribution of the individual data series, we utilized the known impact of ECMO flow on patient oxygenation for an explorative analysis. The first day of support (Day 1) and 2–3 days before weaning (Day x) were chosen as the representative values for a non-parametrical Spearman rank correlation test to reveal an association between Q_ECMO_ and SpO_2_ (first analysis day 1, second analysis day x). We postulated SpO_2_ to be independent of ECMO flow. Based on the different interindividual flow ranges, selecting five evenly distributed ECMO flow values was required from each patient data set for the correlation test. A repeated analysis at two different days should have shown whether the correlation could be confirmed during the entire ECMO period. A *p*-value < 5% was defined as significant. All data were collected as worksheets and analysed with Stata/IC 12.1 (Stata Corp., College Station, TX, USA) and Statistical Package for the Social Sciences (SPSS) 22.0 (IBM Corp., Armonk, NY, USA). Data graphing was carried out with OriginPro 2023 (OriginLab, Additive GmbH, Friedrichsdorf, Germany). To model a correlation between recirculation, gas exchange and patient oxygenation, we applied a curve fitting to the data and included the fitting line as well as its 95% CI in the graphs as appropriate.

## 3. Results

Nineteen patients could be included for assessment of recirculation, VO_2 ECMO_ and SpO_2_. The median age of the patients was 48 years and the median duration of the ECMO support was 12 days. In total, 52% of the patients could be weaned from ECMO and 42% could be discharged. In 70% of cases, patients received a double lumen cannula (Avalon Elite^®^, Maquet KG, Rastatt, Germany), whereas the remaining patients were cannulated through a femoro-jugular access with two separate cannulas.

### 3.1. ECMO Flow and Recirculation

The frequency distribution of all recirculation data showed a median recirculation rate between 14–16% at day 1 and 2–3 days (Day x) before weaning with a maximum rate of approximately 58%. High recirculation rates above 35% were noted in only 13–14% of all measurements (Figure 1). To exemplify the distribution of REC and show the median value we added additional distribution curves (red and grey, day 1 and day x, respectively) to the frequency distribution of REC.

### 3.2. Recirculation and ECMO Oxygenator Transfer

Figure 2 depicts the effect of increasing blood flow on recirculation and ECMO oxygen transfer. With increasing ECMO blood flow and the corresponding increasing recirculation (Figure 2A), VO_2 ECMO_ rose from 50 mL O_2_ × min^−1^ to a maximum value of approximately 220 mL O_2_ × min^−1^ (Figure 2B). Above a plateau at 4 L × min^−1^ ECMO blood flow, VO_2 ECMO_ r reached either a plateau (10 out 19 patients) or showed reduced oxygen transfer (9 out of 19 patients) simultaneous to a further rising recirculation rate.

### 3.3. ECMO Blood Flow, Recirculation, ECMO Oxygen Delivery and Arterial Patient Saturation

In contrast to VO_2 ECMO_ with its mostly plateau reaching values, DO_2 ECMO_ rose linearly as recirculation increased with higher Q_ECMO_. Even with high Q_ECMO_, no decrease was observed (Figure 3B). Analogous to the linear trend of DO_2_, SpO2 increased linearly with increasing Q_ECMO_ and recirculation without any relevant decrease at high Q_ECMO_ (Figure 3C). The binomial test results showed a strong positive relationship between Q_ECMO_ and SpO_2_ on Day 1 and Day x (*p* = 1.51 × 10 ^−6^ and 7.19 × 10 ^−7^, respectively).

### 3.4. Cardiac Output and Its Impact on Recirculation

To unravel the impact of CO on recirculation, we investigated patients with high CO variation with a difference >4 L × min^−1^ between the lowest and highest CO during ECMO. The data was separated into two groups with low and high CO limited by the 0.15 and 0.85 quintiles. A clinically common Q_ECMO_ of 4 L × min^−1^ was chosen as a point of reference for comparison of both groups. Exemplified by one data set (patient #10), higher CO decreased recirculation by between 13% (Figure 4).

## 4. Discussion

Our findings suggest that recirculation is a clinically variable phenomenon that reduces extracorporeal oxygenation depending on ECMO blood flow. Second, higher blood flow seems to not be negatively associated with patient arterial oxygenation. Third, recirculation appears to be considerably influenced by patient CO.

How can these findings be reconciled with the evidence from numerous studies indicating that recirculation causes decreased efficiency with a negative impact on patients?

### 4.1. Recirculation, Gas Transfer and Patient Oxygenation

Recirculation is thought to be associated with a decreasing effective pump flow beyond an optimal flow—Fortenberry reported that effective ECMO flow is delivered to patients’ plateaus after an initial increase and is then reduced with the rising pump flow [6,8]. Additionally, several investigators have assumed that high recirculation may markedly diminish extracorporeal gas exchange and cause patient hypoxemia [5,6,9]. We indeed could confirm with our findings that VO_2 ECMO_ in 9 out of 10 patients decreased beyond a variable optimum flow. Nevertheless, high values of Q_ECMO_ were thought not only to reduce extracorporeal oxygen transfer, but to also cause patient hypoxemia. Consequently, high ECMO flows of approximately 5–6 L × min^−1^, with obviously reduced extracorporeal DO_2_, were prevented in the past [8,10,13]. We adopted this assumption previously, calculated an individual optimum flow and recommended this as a reasonable maximum flow before the study period. Despite this calculated optimum flow, our patients clinically often demonstrated an increased arterial saturation with pump flows higher than the computed optimal flow. Togo reported a similar effect with adult goats—higher flow and recirculation resulted in increasing arterial oxygenation [23]. Nunes further described a positive association between arterial blood oxygen content and increasing ECMO blood flow [16].

Because of the lack of systematic data, it remained open whether increasing ECMO blood flow or recirculation was accompanied with changed or reduced patient oxygenation. To disentangle the clinical paradox of increasing patient oxygenation despite reduced oxygenator function, we then focused on patient parameters. Akin to the pre-trial period, SpO_2_ increased linearly with increasing ECMO blood flow and recirculation in our study. This was reproducible at the beginning, as well as before the weaning of ECMO. Although the patient oxygenation may reach a plateau or even slightly decrease at flows >5–6 L × min^−1^ beyond an “optimum” flow, the ECMO flow dependent extracorporeal DO_2_ seemed clinically to be a determining factor of patient oxygenation. ECMO blood flow did not limit but, rather, ensured and increased patient arterial oxygenation. It could cautiously be assumed that increasing ECMO blood flow is much more important in clinical practice than a reduced oxygenator oxygen transfer which strictly affects oxygenator performance. Our prospective and systematic clinical data confirmed the findings of Zanella, who used retrospective patient data to compute a predictive model: with a high accuracy he could show that higher ECMO blood flows are associated with higher patient arterial oxygenation [17]. Although available as a teaching tool a validation of this predictive model with larger and especially prospective clinical data has not been carried out.

### 4.2. Recirculation and Patient CO

Besides gas exchange, recirculation as a function of CO featured a certain pattern. In patients with a high variation of CO > 4 L × min^−1^ during support recirculation, it was negatively correlated with CO. Higher CO markedly reduced recirculation. This highlights how recirculation is not a fixed but dynamic process that could be misleading if the affecting CO is not considered. Although low recirculation because of high CO may increase the extracorporeal DO_2_, the mismatch between higher CO and lower pump flow still leads to prepulmonary shunting and refractory patient hypoxemia as described by Levy [3]. Therefore, it is important to reassess recirculation during daily practice as it may exhibit major changes.

### 4.3. Clinical Relevance

The estimation of recirculation based on theoretical calculations of the native venous saturation were instruments that allowed for initial insights into the complex interaction between native venous circulation and a technical ECMO system [9,15,24,25]. They permitted a mathematical approximation and resulted in estimated recirculation fractions between 20–30%, which seemed to be clinically prevalent [9]. Several algorithms have been developed for the clinical management and estimation of recirculation phenomena and patient oxygenation [4,26,27]. The existence of several mathematical unknowns in the calculations and a lack of comprehensive clinical validation limited the usability of these methods. The overestimation of ECMO-related technical efficiency further masked the importance of ECMO flow-dependent extracorporeal DO_2_ and its positive impact on patient oxygenation. Quite recently, the focus has shifted from ECMO efficiency towards an optimized extracorporeal DO_2_ with higher oxygenation in animals [23]. A more careful delineation of our data, beyond extracorporeal oxygen transfer, revealed extracorporeal DO_2_ to be independent from extracorporeal gas transfer. The combined use of blood gas analysis and the ultrasound dilution technique fostered our confirmation of recirculation as a clinically prevalent phenomenon with reduced oxygenator efficiency that does not automatically reduce patient arterial oxygenation with higher ECMO blood flows. The positive impact of higher ECMO flow on patient arterial oxygenation is a major contribution to the clinical context, enabling optimization of patient oxygenation.

### 4.4. Limitations

Based on the observational study character, we were not able to vary cannulation and cannula positions, and thus could not draw any conclusions on the impact of cannulation. Additionally, 68% of the patients received a double-lumen cannulation, which further impedes any assessment. Additionally, a predominant double-lumen cannulation further impedes any assessment. Systematic data on varying cannulation, venous volume and altering CO may only be derived with an in vitro model.

Although the validity of pulse contour analysis applied to CO measurement may be questioned, we could exclude any possible right ventricular mixing phenomena with this method and estimate CO without additional invasive methods. More recent results indeed confirmed that mixing effects during ECMO may impact the accuracy of traditional thermodilution based cardiac output measurement [28]. Russ reported in an animal experiment that during ECMO measurement of cardiac output through thermodilution resulted in an overestimation of measured CO compared to aortic blood flow measured directly with an ultrasound flow probe. Even with low blood flows a mean difference approximately 1.38 L × min^−1^ could be observed which increased with higher blood flows and increasing recirculation measured with ultrasound dilution. Flow differences up to 2.7 L × min^−1^ may be assumed to be a high proportion of the cardiac output. The authors concluded that the additional recirculation within the ECMO system caused an additional recirculation signal within the algorithm of CO-calculation and affected the temperature changes which influenced the computed CO values. A higher accuracy during ECMO would require a modification of the current algorithms for the CO measurement. Haller observed an overestimated cardiac output up to a maximum of 300% with conventional thermodilution during ECMO and recommended using dye dilution techniques instead, which were obviously not influenced by recirculation kinetics [21]. The used pulse contour analysis used in our study is classified as an internally calibrated system based on biometric data and pulse wave characteristics depending on vascular resistance and compliance [29]. Depending on vascular tone changes, Slagt reported error rates below 30% with the internally calibrated pulse contour analysis in normo- and hypodynamic conditions and classified this system to be sufficiently accurate [30]. Bond reported a more recent pulse contour analysis during ECMO to be in good agreement to echocardiographic cardiac output measurement with a mean bias of −0.2 L × min^−1^ and a percentage error of 24% [31]. We, therefore, assume the used pulse contour analysis to be a compromise in estimating cardiac output. Regarding patient oxygenation, traditional blood gas analysis would have resulted in more precise data of the patient gas exchange but would have also caused extra blood loss. Therefore, pulsoxymetry as a surrogate parameter was a justifiable compromise with this trial.

## 5. Conclusions

The measurement of recirculation as a prevalent clinical phenomenon through the ultrasound dilution technique is a real-time method with higher precision than mathematical approximations. The systematic and direct clinical measurement confirmed empirical ranges of recirculation approximately 10–30%, which reduced extracorporeal oxygen transfer. However, higher ECMO flows allowed increased patient arterial oxygenation, and thus, may prevent refractory hypoxemia. To be of true diagnostic value, recirculation should always be considered multifactorial. The ECMO blood flow itself, its ratio to cardiac output and the cannula position may intricately affect patient oxygenation and should be carefully considered.

## Figures and Tables

**Figure 1 jcm-12-00416-f001:**
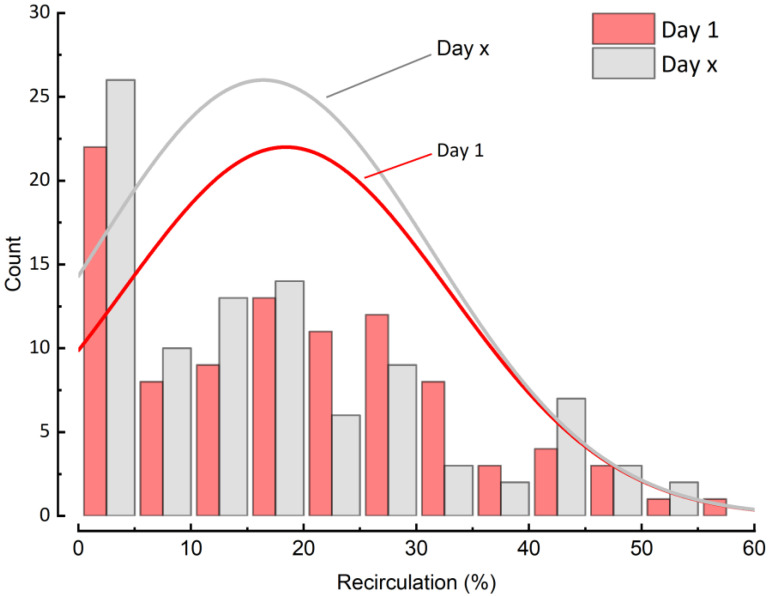
Distribution of recirculation at Day 1 and 2–3 days before weaning.

**Figure 2 jcm-12-00416-f002:**
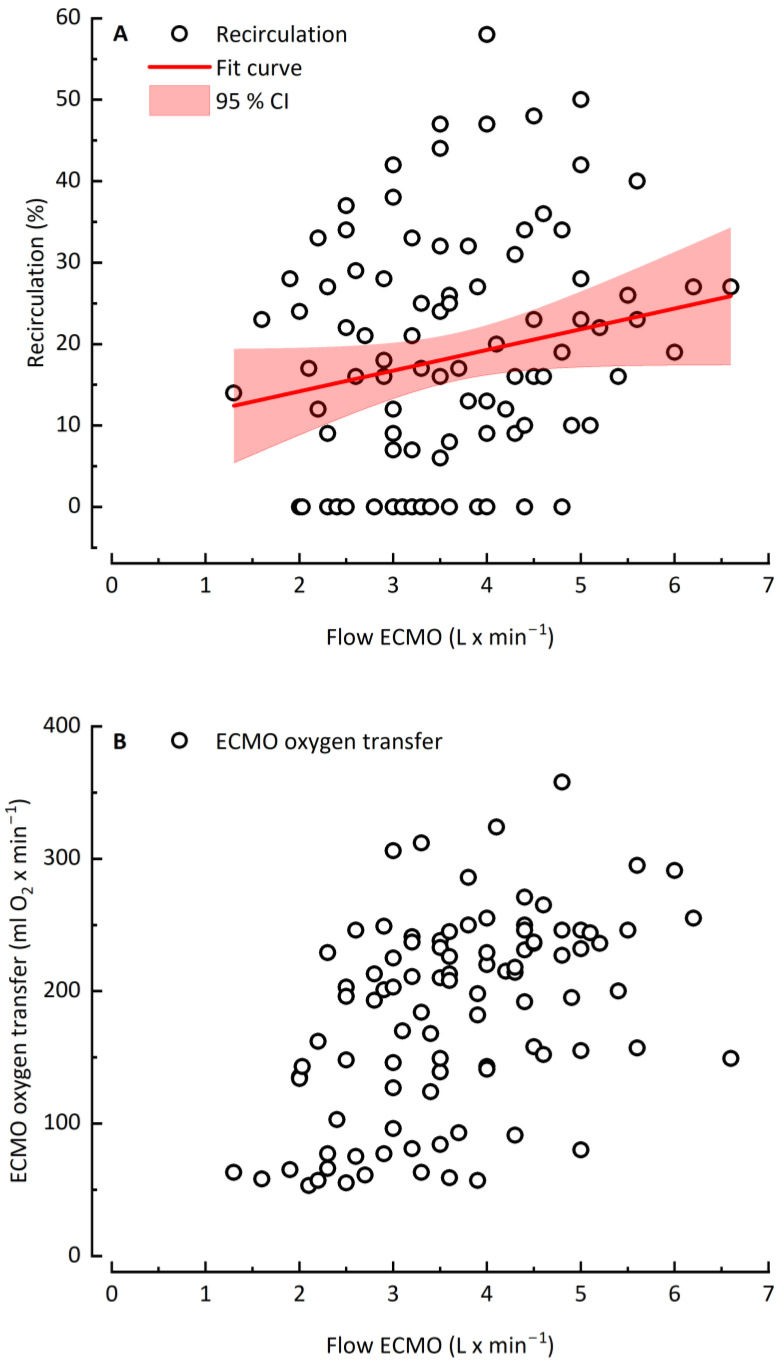
Recirculation rate (**A**) and ECMO oxygen transfer (**B**) as a function of ECMO flow rate.

**Figure 3 jcm-12-00416-f003:**
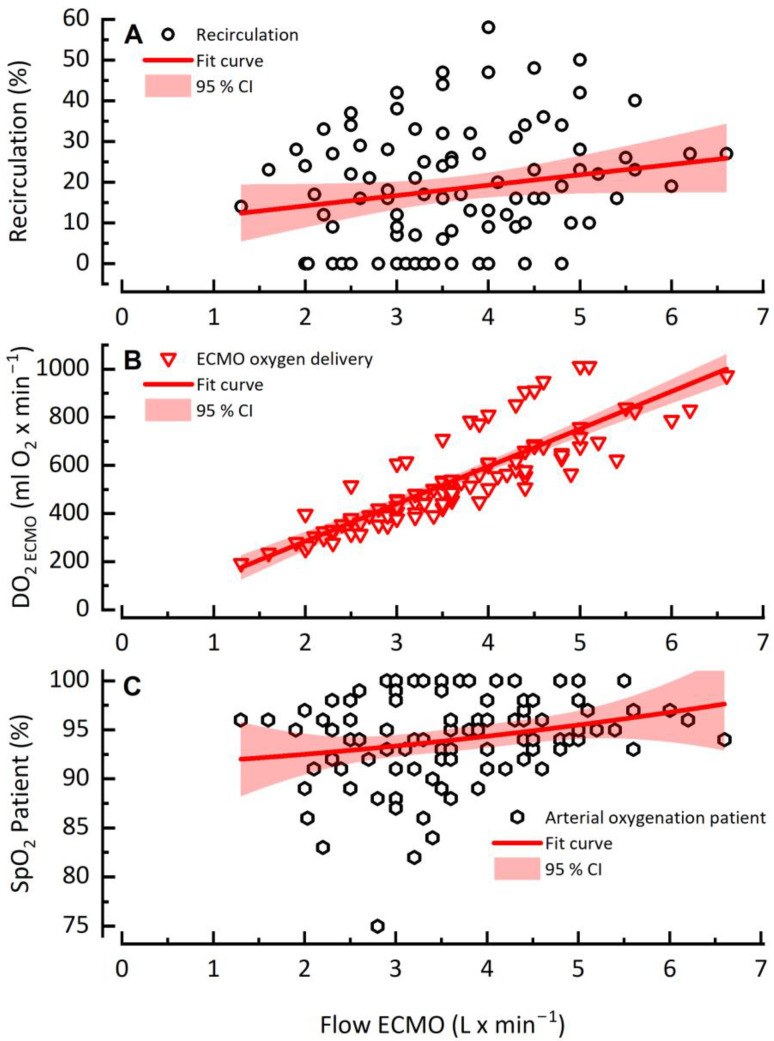
Recirculation rate (**A**), DO_2 ECMO_ (**B**) and patient arterial saturation (**C**) as a function of ECMO flow rate.

**Figure 4 jcm-12-00416-f004:**
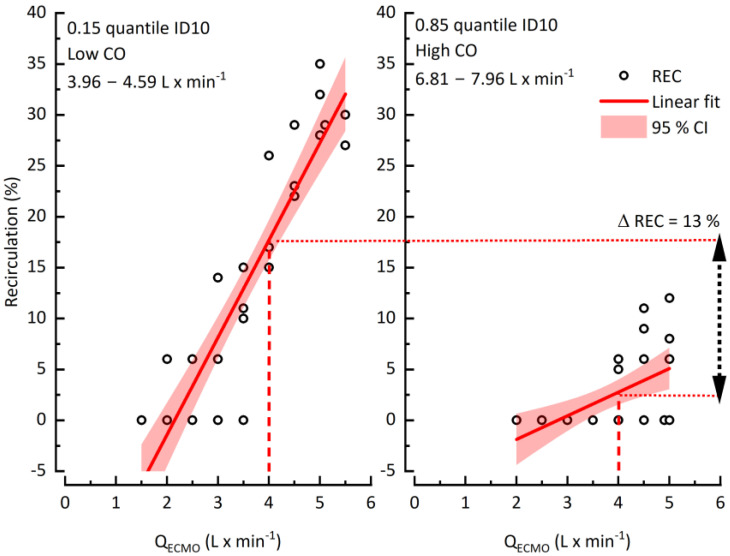
The impact of cardiac output on recirculation.

## Data Availability

The data presented in this study are available on request from the corresponding author.

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
