# Peer review of "The Impact of Recirculation on Extracorporeal Gas Exchange and Patient Oxygenation during Veno-Venous Extracorporeal Membrane Oxygenation—Results of an Observational Clinical Trial"

_jcm, 2023, doi:10.3390/jcm12020416_

Round 1

Reviewer 1 Report

In this observational study, the Authors described the relationship between ECMO recirculation, cardiac output, oxygen delivery and consumption.

The topic is relevant for clinicians who works with ECMO, as recirculation may significantly impair the extracorporeal oxygen delivery. The thermodilution technique is an interesting method to assess recirculation during ECMO.

I have some comments:

Background

-       This topic has already been widely studied. What is the new research question which the authors aim to investigate? This should be emphasized.

-       The background is not properly described, and some parts of the introduction are confusing. “Besides the proximity of the cannulas, which primarily determines the recirculation fraction (REC),  the ratio of ECMO flow (QECMO) and native CO may induce relevant venous shunting with diminished DO2 if CO is higher than QECMO”. These are two separate concepts. REC increases when cannulae are close and when the CO/Qecmo ratio is low. Contrarily, when the CO/Qecmo is high, only a smaller fraction of venous blood is oxygenated by ECMO (I would not use “venous shunting”)  resulting in a systemic hypoxia in patients with high intrapulmonary shunt. I would soggest a major revision of the introduction section, with very clear and concise concepts.

Methods

-       “The primary outcome was the maximization of patient oxygenation.” This is not an outcome. Outcomes may be, for ex:

Ø  to describe the % of REC in a population of vvECMO

Ø  To explore the impact of REC on oxygenation (or ecmo VO2) at increasing Qecmo

Please edit primary and secondary outcomes accordingly.

-       I do not understand the meaning/relevance of DO2 eff, while I agree that patient DO2 (CartO2 x CO), ECMO Qeff and ECMO VO2 are important parameters. Is DO2 eff calculated on ECMO blood flow or patient CO? ECMO DO2 is irrelevant, whereas patient DO2 is a variable which does not require any correction. If two patients have a DO2 of 1000ml O2/min the DO2 is the same, regardless their REC. I suggest to remove DO2eff from the paper.

-       I suggest to rename VO2 > VO2 ECMO or similar to clarify that you are not talking about lung VO2 (Cart-Cven)xCO.

Results

-       Figure 1. What do the two red and grey curves indicate?

-       Section 2.a: “Figure 2 depicts the impact of recirculation on gas transfer.” This is not true. Figure 2 depicts the effect of increasing BF on recirculation and ECMO VO2.

-       “With increasing recirculation (Fig. 2A), VO2 rose from 50 ml O2 x min-1 to a maximum value of approximately 220 ml O2 x min-1” Same as above. Did you mean with increasing ECMO BF? Please check this type of (major) error throughout the paper

-       Figure 2b shows a polynomial relationship where VO2 decrease over a BF of 5L min. This is unlikely. At very high blood flows the recirculation increases, and Qeff reaches a plateau (but does not decrease)

-       Please spell out abbreviation (e.g. DO2) in the figure legends

-       If you want to study the effect of recirculation on other parameters (e.g. DO2), the former should be your X-axis in the graphs.

Discussion

-       Our findings suggest that, first of all, recirculation is a clinically variable phenomenon that reduces extracorporeal gas exchange as well as flow and DO2 to patients depending on ECMO flow.” I would say REC reduces -extracorporeal oxygenation-, not gas exchange (you did not explore CO2 which is not significantly affected by recirculation). I do not think your data support the fact that REC reduces DO2.

-       Please modify the discussion based on the modification suggested above.

-       Uncalibrated pulse contour methods lack in precision/accuracy. I understand that other techniques (TD) suffer from a relevant impact of BF, but the limitation of the chosen CO measurement method should be discussed more widely.

-       I would discuss that the impact of recirculation on oxygenation in patients on vvECMO can be computed in-vitro (ref. Zanella 2016 JoCC A mathematical model..)

Author Response

Dear Reviewer,

Thank you for your valuable and interesting comments and for giving us the opportunity to improve and resubmit our manuscript “The Impact of Recirculation on Extracorporeal Gas Exchange and Patient Oxygenation during Veno-venous Extracorporeal Membrane Oxygenation – Results of Observational Clinical Trial”.

The manuscript has been revised according to the comments raised by the reviewer to the best of our ability. Please find a detailed reply to the reviewer comments attached with this revision.

We would like to thank the reviewer for the constructive and competent criticism, and we hope that our manuscript will be acceptable for publication in Journal of Clinical Medicine.

We absolutely agree with you that numerous studies had been carried out regarding the complex phenomenon of recirculation, its affecting determinants and the corresponding arterial oxygenation. Most of them are done mathematically which did not measure but estimate recirculation. As recirculation is partially fixed depending on the cannula position it seems relatively simple to calculate it. The further interdependence between recirculation and cardiac output is an essential issue which complicates this calculation. High cardiac output may reduce recirculation but also reduce arterial oxygenation, lower cardiac output may increase recirculation but probably increase arterial oxygenation. These two input conditions allows not to use a mathematical solution but would only allow a mathematical approximation. This was a major limitation of all the mathematical studies in the past and limited the applicability for a clinician. One of the central points surrounding recirculation was the efficiency of the ECMO and its impact on arterial oxygenation. Sreenan (the only reference in Zanella’s paper which described recirculation) stated: „Recirculation is a frequent problem during VV ECMO and leads to less efficient oxygen delivery. If recirculation becomes excessive, effective flow and thus oxygenation of systemic blood will be reduced. Increasing flows further to try and remedy the situation paradoxically can lead to further recirculation and a further decrease in oxygenation“ Especially the last part is an important issue which influenced the international ECMO community.  The central message from Sreenan was translated into the red book from ELSO. One central graph depicted the increasing inefficiency with increasing blood flow. This influenced the ECMO community towards a low flow regime with obviously less recirculation but also low arterial oxygenation. 

Our group regularly observed an opposite clinical scenario: higher blood flow led to higher arterial oxygenation. Therefore we initiated our systematic clinical observation with direct measurement of recirculation and not with mathematical approximation. To our knowledge our clinical study is the only one which systematically and prospectively measured recirculation. We think that prospective systematic measured data is superior to approximations (even with excellent agreement to retrospective clinical date like with Zanella)

We could confirm Zanella's approximation with prospective clinical data: higher blood flow is associated with higher arterial oxygenation. This is a marked contrast to the established low flow regime and the efficiency debate described above. Zanella used retrospective data from patients and included estimated values of recirculation and thus used the same approach as all the other mathematical approximations which tried to recalculate recirculation based on numerous mathematical unknowns. We can not solve equations with more than one interdependent variables. Although Zanella could use cardiac output data, which validity is still under debate, he unfortunately only mentioned low cardiac output associated with higher recirculation. We could confirm this with our study as we measured one female patient with a high grade tricuspid valve insufficiency who showed high recirculation values above normal. Zanella unfortunately did not mentioned higher cardiac output which may markedly reduce recirculation but also cause lower arterial oxygenation. Thus he obviously neglected the twofold impact of cardiac output on recirculation. How should an equation than be solved with embedded estimated values with a complex interdependence to completely different clinical scenarios.  

Our group investigated this unsolved situation and carried out an experimental setup with a vena cava model where we could vary the cannula position as well as the cardiac output, the blood flow and thus the ratio of the latter two variables. We could show that cardiac output is a major determinant of recirculation and may show completely different values depending on the ratio of cardiac output and blood flow (unpublished date but available as a thesis in german language). Our experimental results confirmed our assumption from our clinical trial which also showed the impact of cardiac output on recirculation. A nominal REC value around 20-30 % with a "normal" CO/BF ratio diminished completely with higher CO and increased up to 50-60 % with lower CO.

Thus we think that clinical measurement has a higher validity than mathematical solutions which may allow computing and ease evaluating patients at bedside. We assume that our observational and prospective clinical study provides the only comprehensive data set about recirculation measurement and its possible clinical consequences which never had been measured in such an extent before. We think this is the novelty of our study. 

Below we replied point by point to your comments:

  1. Background: 
    #Recirculation has widely been studied
    Reply: T
    he phenomenon of recirculation has indeed been widely studied, numerous studies tried to solve the intricately connected processes with mathematical, experimental or computed approach. The majority of them had in common that they only estimated recirculation, depending on several other unknowns (which we described in the introduction) None of them carried out a systematic clinical investigation with direct measurement. Thus we think that our trial adds new knowledge to this debate.
    #Concepts not properly described
    Reply: we revised the introduction section, excluded the confusing passages and substantiated the description of recirculation, its impact and our research question
  2. Methods:
    #Primary and secondary outcomes
    Reply: We interestingly had the same discussion with our statistician. We as clinicians suggested to describe the outcome as the impact of REC on oxygenation (as you suggested) Our statistician suggested to use maximization of patient oxygenation. For a better readability we changed the primary outcome to the impact of REC on patient oxygenation and ECMO VO2 at increasing Qecmo. Secondary endpoints were also modified.
    DO2 eff: we removed DO2 eff from the paper, but we still think that effective ECMO flow or oxygenator related oxygen delivery is an important parameter like a thumb rule to explain the effect of recirculation on patient oxygenation. Let's assume a low or CO/Qecmo ratio of 6/4 l/min. With an often experienced REC around 20-30 % a high proportion of blood is returned to the ECMO and does not contribute to the delivered oxygen to the patient irrespective of the patient DO2. The reduced ECMO flow (4 - 20-30 %) may then be further calculated into the ECMO related DO2 eff.
    However, as this may be confusing for the reader we removed DO2 eff. 
    VO2: VO2 was renamed as VO2 ecmo.

  3. Results:

    #Figure 1, gray and red lines
    Reply: The gray and red lines indicate the distribution curve of REC during day 1 and day x and should illustrate the median values. A description was added to the text.

    #Figure 2a:
    Reply: We corrected figure 2 regarding the impact of ECMO blood flow on REC and ECMO VO2 and revised the corresponding text.

    #Figure 2b, polynomial relationship
    Reply: We interestingly observed exactly the phenomenon of a decreasing ECMO VO2 with increasing blood flow: 9 out of 19 patient showed a polynomial U-shaped pattern, 10 out of 19 patients showed obviously a plateau with higher blood flows (which was your assumption). Using a polynomial fit thus may not be completely justified. We removed the polynomial fit and presented only the raw data in figure 2a/2b. Please see the attachment where the  pdf-file "authorcoverletter-25243653..." (it was not possible to rename the file) shows the single VO2 pattern of all 19 patients. We decided to present only summarized data, as 19 single graphs would be difficult to publish and be very small. I would suggest to provide the graph as supplementary material

    #Figure 2, abbreviations
    Reply: The former abbreviations has been changed into spelled out phrases in the figure legend.

    #Figure 3, DO2 eff
    Reply: Analog to the methods section we removed the DO2 eff from the figures and thus eased the readability. We still think that REC reduces ECMO oxygen delivery and may be used as a thumb rule to explain the consequences of high REC rates. With a nominal REC around 20-25 % and a blood flow around 5 l/min only 5 l/min - 20-25 % contributes to the oxygen delivery (irrespective of the patients CO). 5 - 20-25% means only 4 l/min blood flow which contributes to DO2 and also changes the BF/CO ratio. However, with removing the DO2 eff we are more cautious with our assumptions.

  4. Discussion

    #REC and DO2
    Reply: We have modified the beginning of our discussion and have been more cautious with our assumptions regarding blood flow, REC and pateint oxygenation. It may be right that increasing REC only reduces extracorporeal oxygenation (VO2 ecmo); this may be explained with a rising oxygenator inlet saturation and thus lower oxygen transfer. Although DO2 is normally independent from VO2 extracorporeal blood flow maybe reduced by REC as a certain proportion returns into the ECMO and does not contribute to the mixing with patient venous blood. This alters the CO/BF ratio and reduces the total DO2 (extracorporeal and patient). Otherwise a low CO/BF may not cause refractory hypoxia. Because of the REC fraction Levy (Ref #3) recommended to adjust the blood flow higher than the patient CO to compensate for the REC fraction.
    #VCO2
    Reply: During our study period we presented preliminary results about BF, REC and patient oxygenation. One criticism was that we did not account for CO2 transfer changes. Until that time no one had clinically investigated such aspects. We made an amendment to our ethical committee and adopted VCO2 into the study plan. Unfortunately we only had limited time and could only include five patients. We observerd Interestingly the same pattern as with oxygenation, with higher blood flows extracorporeal CO2 transfer was reduced, the patient paCO2 remained stable despite higher flows and showed no decrease. Because of this small sample number we were very cautious to draw any conclusion and did not include the results in this manuscript. However, the findings were extensivly discussed and we were congratulated for our findings.
    #Thermodilution
    Reply: We have extended the discussion surrounding the validity of cardiac output measurement. Even uncalibrated pulse wave analysis methods seems to have a sufficient accuracy compared to conventional thermodilution.
    #computing impact of recirculation
    Reply: We included a short discussion of the findings of Zanella and the difference to our study.

    Johannes Gehron

Reviewer 2 Report

Comments to the author:

The manuscript entitled “The Impact of Recirculation on Extracorporeal Gas Exchange and Patient Oxygenation during Veno-venous Extracorporeal Membrane Oxygenation – Results of An Observational Clinical trial.” This study analyzed that increasing DO2 is more important than reducing VO2. A negative correlation of recirculation to CO adds to the complexity of this phenomenon. Patient oxygenation may be optimized with direct measurement of recirculation. It needs some revisions before acceptance.

Comments

The authors have taken less number of subjects. Moreover, cardiopulmonary  exercise tests (CPET) need to measure the given subjects.

Author Response

Dear Reviewer,

I would like to thank you very much for your critical comments on our manuscript and I would like respond to them:

  1. Language and style: the whole manuscript has been checked twice by an renowned international language editing service before submission. However, we are pleased to send the revised manuscript again to the editing service.
  2. Number of subjects: before starting our trial we calculated the sample size with out department of statistics. Due to the runtime of the ECMO (days up to several weeks), the daily measurement of the patients and the comparison of patient data between one of the first and one one the last days we yielded several hundred data point for analysis. The dept. of statistic assessed the data and the number of patient to be sufficient for analysis. We assumed to have a sufficient and valid number of patients.
  3. Cardiopulmonary exercise testing: recirculation is a phenomenon which occurs only during extracorporeal support and which describes the interaction between the extracorporeal blood flow, the cannula position and the native cardiac output. Changes in the arterial oxygenation may be due to recirculation or shunting during ECMO and not due to reduced exercise capabilities of the patients. CPET may be used as a measure post ECMO to reveal reduced exercise capabilities. With our trial we focused on intra-ECMO parameters which were depending on the recirculation and would not appear post-ECMO. Therefore we made no follow-up, did not use CPET and assume our findings to be independent form exercise testing.

on behalf of the authors

Johannes Gehron

Round 2

Reviewer 1 Report

I want to thank and congratulate the Authors for their detailed and extensive response letter. I believe that after this revision the paper is more solid and its conclusions are more sustained by the study findings. I understand the Authors point about DO2eff, but I believe it was quite misleading for the reader, whereas the ecmoVO2 and Qeff well describe the effective oxygen and flow delivered by the extracorporeal support.

I have no further comments 

Thanks again for the privilege of reviewing this article 

Reviewer 2 Report

It can be accepted in present form